# Impact of Chitosan-Genipin Films on Volatile Profile of Wine along Storage

**M. Angélica M. Rocha [1], Manuel A. Coimbra [1] , Sílvia M. Rocha [1] and Cláudia Nunes [2],\***

[1] LAQV-REQUIMTE, Department of Chemistry, University of Aveiro, 3810-193 Aveiro, Portugal; angelicamartins@ua.pt (M.A.M.R.); mac@ua.pt (M.A.C.); smrocha@ua.pt (S.M.R.)
[2] CICECO, Department of Chemistry, University of Aveiro, 3810-193 Aveiro, Portugal
\* Correspondence: claudianunes@ua.pt; Tel.: +351-234-372581; Fax: +351-234-370084

**Featured Application: New opportunities for winemaking industries are open by the use of chitosan-genipin films as technological adjuvant for white wine preservation to replace sulfur dioxide, maintaining their varietal key odorants and organoleptic characteristics.**

**Abstract:** Chitosan-genipin films have been proposed for preservation of white wine, maintaining their varietal key odorants and organoleptic characteristics of sulfur dioxide treated wines. Nevertheless, these wines showed aroma notes that slightly distinguish them. It is possible that during the contact of films with wine for at least 2 months, after fermentation and prior to bottling, interactions or chemical reactions are promoted. In this work, wine model solutions with volatile compounds in contact with chitosan-genipin films were performed to evaluate their evolution along time. To complement these analyses, the volatile compounds of white and red wines kept in contact with chitosan-genipin films during 2 and 8 months were also studied. The results obtained allowed us to conclude that the contact of chitosan-genipin films with both white and red wines tend to retain long carbon chain volatile compounds, such as ethyl hexanoate and octan-3-one. It also promoted the formation of Maillard reaction products, such as furfural by dehydration of pentoses and Strecker aldehydes, such as 3-methylbutanal and phenylacetaldehyde, by degradation of amino acids. This study reveals that the use of chitosan-genipin films for wine preservation is also able to promote the formation of compounds that can modulate the wines aroma, maintaining the varietal notes.

**Keywords:** chitosan; genipin; Maillard reaction; Strecker degradation; hydrophobic interaction

## 1. Introduction

Chitosan has legal application as clarifying agent in wine industry [1–4]. However, several other different applications [5] have been proposed exploiting its properties, as complexing [6,7], preservative [8,9], and encapsulating agent [10,11] in wines, as in beverages in general [12].

For wine, the chitosan from *Aspergillus niger*, was authorised by International Organisation of Vine and Wine, with a maximum concentration of 1 g/L [13], marketed as Oneobrett® and Bactiless™ brands. Depending on the different purposes, such as to control the population of micro-organisms, in special *Brettanomyces* [14,15] and for uptake of ochratoxin A [6,7], concentrations up to 5 g/L have been admitted by the European Union [16]. Chitosan is able to decrease the amount of sulphur dioxide-bound and, in this way, increase the microbial activity of sulphur dioxide in red wine, as well as its colour stability [17]. When added to pre-fermented wines, chitosan can influence their volatile composition, dependent on yeast used for fermentation, resulting in wines with different characteristics [18]. The reticulation of chitosan with genipin, forming acidic aqueous resistant chitosan-genipin (Ch-Ge) films [19], has been proposed for white wine preservation due to metal complexation properties [20], which is on the basis of the antimicrobial and anti-oxidant activities of the films [21]. This reticulation allows the use of chitosan from

different origins than fungi, including seafood because the allergic response triggered by tropomyosin [22] does not occur, allowing its use as a technological adjuvant in wines that can be taken by seafood allergic patients [23].

The organoleptic characteristics of colour, taste, and aroma of white wines treated with Ch-Ge films are improved when compared to the same wines prepared using sulfur dioxide as preservative [21]. These wines were rated by an expert sensory panel with positive notes attributed to the increase in benzaldehyde and furfural possibly formed due to the reaction between amino groups and sugars in the presence of the films [21]. The improved organoleptic characteristics of these wines have also been related to the decrease in heptan-2-one and nonan-2-one, with buttery and fatty odours, and 4-ethylphenol and 4-ethylguaiacol, with clove and spicy odours [21]. Chitosan has been shown to have capacity to retain ketones [24], as well as volatile phenols [25,26].

The present work intends to identify the type of reactions and interactions that can be promoted by Ch-Ge films with impact on the volatile composition of white and red wines in contact with the films during 2 and 8 months. For this, wine model solutions containing Ch-Ge films were prepared to study the possible mechanisms of reaction of the films with arabinose, phenylalanine, or leucine. In addition, wine model solutions with individual compounds in contact with Ch-Ge films were also used to relate the possible retention properties with structural characteristics and functional groups of volatile compounds. The analysis of the white and red wines volatile composition when in contact with Ch-Ge films were revisited, expanded, and interpreted to validate the observations obtained by model solutions.

## 2. Materials and Methods

### 2.1. Chemical Reagents and Standards

Chitosan of medium molecular weight (150 kDa) from shrimp (CAS#9012-76-4) was used. It was 88% deacetylated and had a purity of 95%, determined as described by Nunes et al. [21]. This polysaccharide was composed by 6.2% of water-soluble material with molecular weight lower than 12–14 kDa (determined by dialysis) and 17% of alkali soluble material (determined by ammonium precipitation). Chitosan and glycerol (99.5%, CAS#56-81-5) were provided by Sigma-Aldrich (St. Louis, MO, USA). Genipin (≥99.0%, CAS#6902-77-8) was supplied from Challenge Bioproducts Co. (Taiwan, China). The films of chitosan crosslinked with genipin were prepared by solvent casting using 1.5% ($w/v$) chitosan in 0.1 M acetic acid, 0.05% ($w/v$) genipin, and glycerol as plasticiser, has reported by Nunes et al. [21].

The standards used in model solutions were purchased from 4 different suppliers: benzaldehyde (CAS#100-52-7, ≥99%) provided by Fluka, Steinhem, Germany; arabinose (CAS#28697-53-2, ≥99%), furfural (CAS#98-01-1, ≥98%), phenylacetaldehyde (CAS#122-78-1, ≥98%), phenylalanine (CAS#673-06-3, ≥99%), and octan-3-one (CAS#106-68-3, ≥98%), provided by Sigma-Aldrich, St. Louis, MO, USA; ethyl acetate (CAS#141-78-6, 99.9%) provided by VWR, France; ethyl hexanoate (CAS#123-66-0, 99%), hexanoic acid (CAS#142-62-1, 99.5%), hexanal (CAS#66-25-1, 99%), hexanol (CAS#111-27-3, 99%), leucine (CAS# 61-90-5, >99%), and 3-methylbutanal (CAS#590-86-3, 97%) provided by Aldrich-Chemie, Steinhem, Germany. All other reagents (acetic acid, ethanol, sodium chloride, sodium hydroxide, and tartaric acid) used were analytical grade.

The SPME holder, for manual sampling, included a fused silica fibre coating with divinylbenzene/carboxen/poly(dimethylsiloxane) (DVB/CAR/PDMS) with 50/30 μm of thickness and 1 cm of length, was supplied from Supelco (Bellefonte, PA, USA). Prior to initial use, SPME fibre was conditioned for 60 min at 270 °C in the gas chromatography injector as recommended by manufacturer. The retention index probe (n-alkane series of C8 to C20 straight-chain alkanes, in hexane) was purchase by Fluka (Buchs, Switzerrland).

## 2.2. Wine Model Solutions

Wine model solutions containing individual compounds (Table 1) were prepared in 9 mL of sealed vials with 10% (*v/v*) ethanol and 0.5% (*w/v*) tartaric acid with pH 3.5 adjusted with 16% (*w/v*) sodium hydroxide, in a total of 3 mL. After immersion of 2 cm$^2$ of Ch-Ge films, the pH was adjusted to 3.5 with 50 μL of 7.5% (*w/v*) tartaric acid. To evaluate the influence of the proportion of film to the analyte, ethyl hexanoate, and hexanoic acid, experiments were performed also using 0.5 and 1 cm$^2$ of Ch-Ge films for the same volume of solution. Reference samples were prepared without Ch-Ge film. All the solutions were placed in the dark, at room temperature, under magnetic stirring. Analyses were performed in different times (Table 1), using three independent vials.

**Table 1.** Composition of wine model solutions in contact with chitosan-genipin films, sampling time, and compounds physico-chemical parameters: molecular weight (MW, g/mol), logarithm of the octanol/water partition coefficient (Log P), and boiling point (BP, °C).

| Standard | C$_{vial}$ (mg/L) | Sampling Time | MW | Log P | BP (°C) |
|---|---|---|---|---|---|
| Hexanol | 2.4 | 20, 80, 140, 260 min | 102.17 | 2.03 | 158 |
| Octan-3-one | 0.058 | 20, 80, 140, 260 min | 128.21 | 2.30 | 168 |
| Hexanal | 17.8 | 20, 80, 140, 260 min | 100.16 | 1.78 | 130 |
| Benzaldehyde | 1.0 | 20, 80, 140, 260 min | 106.12 | 1.48 | 179 |
| Ethyl acetate | 9.6 | 20, 80, 140, 260 min | 88.11 | 0.73 | 77 |
| Ethyl hexanoate | 0.11 | 20, 50, 80, 140, 200, 260 min | 144.21 | 2.83 | 167 |
| Hexanoic acid | 10.9 | 20, 50, 80, 140, 200, 260 min | 116.16 | 1.92 | 205 |
| Arabinose | 1.5 | 0, 34, 41, 49, 106, 120 days | 150.13 | | |
| Leucine | 1.31 | 0, 30, 60, 74, 91, 120 days | 131.17 | | |
| Phenylalanine | 1.65 | 0, 34, 41, 49, 106, 120 days | 165.16 | | |

For the study of retention of volatile compounds by Ch-Ge films, each one of the 7 compounds under study was assayed after 20, 80, 140, and 260 min, in a total of 168 assays. For the study of promotion of Maillard reaction by Ch-Ge films, each one of the 3 compounds (arabinose, leucine, and phenylalanine) under study were assayed in five different times along 120 days. Control solutions containing genipin (74 mg/L) or glycerol (1.7 g/L) in contact with arabinose and phenylalanine were also assayed in triplicate at 36 and 120 days.

## 2.3. Wine Samples

White wines were produced using Encruzado (*Vitis vinifera* L.) grapes variety, while red wine production used Touriga Nacional (*Vitis vinifera* L.) grapes variety from Dão Appellation of 2010 harvest. The winemaking of white (13.1 alcohol, *v/v*) and red (14.4 alcohol, *v/v*) wines was performed according to Nunes et al. [21] and Santos et al. [27], respectively. After alcoholic and malolactic fermentation, for white and red wines, respectively, the chitosan-genipin (Ch-Ge) film was added in a proportion of 100 cm$^2$ of film per bottle (3 bottles of 750 mL for each variety) of wine. Both wine samples were produced by Dão Sul SA (Carregal do Sal, Portugal) and stored during 2 and 8 months in dark at 80% relative humidity and a temperature ranging from 10 °C to 15 °C. These wines were part of a set of white and red wines prepared to study the feasibility of Ch-Ge films [21] and high hydrostatic pressure [27] to replace sulphur dioxide as wine preservative after fermentation. These wines were never used in any previous publication.

## 2.4. Analysis of Volatile Compounds

### 2.4.1. Extraction of Volatile Compounds

The volatile compounds were extracted using headspace solid phase micro-extraction (HS-SPME) according with the methodology described by Nunes et al. [21]. Briefly, 3 mL of the samples (wine model solutions and white and red wines) and 0.6 g of sodium chloride were placed into vial (9 mL) and immersed in a thermostated bath at 40.0 ± 0.1 °C. Then, SPME fibre was inserted into the headspace along 20 min, under continuous stirring at

400 rpm, to cause the extraction and concentration to occur in the same step. The assays were performed in triplicate.

### 2.4.2. Determination of Volatile Compounds in Wine Model Solutions

The content of each volatile compound in the wine model solutions was determined by HS-SPME combined with gas chromatography using a flame ionisation detector (GC-FID). After the extraction and concentration step, the SPME coating fibre was manually introduced into the GC-FID injection port (220 °C). The GC was equipped with 30 m column DB-FFAP (J&W Scientific, Folsom, CA, USA) with 0.25 mm internal diameter (i.d.) and 0.32 μm thickness. The separation of each volatile compound, namely benzaldehyde, furfural, hexanol, hexanal, and phenylacetaldehyde, was carried out with the oven temperature program reported by Nunes et al. [21]. The separation of octan-3-one was done using the following oven temperature program: initial temperature 35 °C during 3 min, an increase in temperature with a rate of 45 °C/min until 220 °C, maintaining for 1 min. The separation of ethyl hexanoate and hexanoic acid occurred with the following temperature program: initial temperature at 40 °C, an increase in temperature at a rate of 3 °C/min until 120 °C, followed by a rate of 40 °C/min until 250 °C and maintain at this temperature for 1 min. In all separations, the flow rate of the hydrogen ($H_2$) was 1.7 mL/min and the detector temperature was set at 230 °C.

The quantitative analysis was obtained by interpolating the peak area on the external calibration curves made with five different concentrations for each standard. The retention of the volatile compounds (benzaldehyde, ethyl acetate, ethyl hexanoate, hexanoic acid, hexanal, hexanol, and octan-3-one) by Ch-Ge films was expressed based on the decrease in content in the vapour phase in comparison with the reference (without films). The formation of volatile products of Maillard reaction (benzaldehyde, furfural, 3-methylbutanal, and phenylacetaldehyde) by Ch-Ge films was expressed based on their content in the vapour phase in comparison with the reference (without films).

### 2.4.3. Determination of Volatile Compounds in Wines

The analyses of volatile compounds of white and red wines were performed by HS-SPME combined with comprehensive two-dimensional gas chromatography with time-of-flight mass spectrometry (GC × GC-ToFMS). A gas chromatograph, LECO Pegasus 4D (Leco, St. Joseph, MI, USA) GC × GC system consisted of an Agilent GC 7890 gas chromatograph (Agilent Technologies, Inc., Wilmington, DE) with a dual stage jet cryogenic modulator (licensed from Zoek) coupled to mass spectrometer equipped with a time-of-flight (ToFMS, 5976 N MS (Agilent Technologies) analyser. The volatile compounds were separated through the combination of two columns: HP-5 (30 m × 0.32 mm i.d. × 0.25 μm film thickness) in the first-dimension ($^1$D)) and DB-FFAP (0.79 m × 0.25 mm i.d. × 0.25 μm film thickness in the second-dimension ($^2$D)). Both columns purchased from J&W Scientific (Folsom, CA, USA).

The chromatographic conditions were the same as reported by Santos et al. [27], such as the injector temperature (250 °C), the temperature program for primary (40 °C (1 min) to 230 °C (2 min) at 10 °C/min) and secondary (70 °C (1 min) to 250 °C (3 min) at 10 °C/min) oven, using helium (2.5 mL/min) as carrier gas. The modulator was offset by +20 °C above in relation to primary oven and the modulation time was 5 s. The mass spectrum parameters were: the transfer line temperature at 250 °C, detector voltage of −1786 V, at 70 eV with ion source temperature at 250 °C, mass range of 33 to 350 $m/z$, and acquisition rate of 125 spectra s$^{-1}$. Total ion current chromatograms (TIC) were processed using data processing software ChromaTOF® (LECO).

For the identification of the compounds, the first criteria is the co-injection of standards (when available). However, it is possible to identify the compounds even in the absence of a standard that consists in a combination of three criteria, namely mass spectrum similarity (provided by commercial libraries Wiley 275 and US National Institute of Science and Technology (NIST) V. 2.0, Mainlib and Replib), relation of calculated linear RIcal (according to

the van Den Dool and Kratz equation [28]) with reported in literature (RIlit) for 5% phenyl-methylsiloxane (or equivalent column), and the presence of a structured chromatogram (in which similar structures are located in similar spaces due to orthogonal separation mechanism that provided the combination of a HP-5 with a DB-FFAP columns) [29]. The use of GC × GC peak area, expressed as arbitrary units (a.u.), allowed to estimate the relative content of each volatile component in wine.

### 2.5. Data Analysis

GC × GC peak areas of volatile compounds were manually extracted from the chromatograms, allowing to build two full data matrices, one for white wine (177 volatile compounds) and another for red wine (206 volatile compounds), at 2 and 8 months of storage and 3 replicates. Two heatmaps were constructed using autoscaled data through Metaboanalyst 4.0 (web software, The Metabolomics Innovation Centre (TMIC), Edmonton, AB, Canada) [30]. Principal component analysis, an unsupervised method, was carried out to reduce the dimensionality of the data matrix maintaining the maximum variability in the volatile composition and to evaluate the impact of chitosan-genipin films on wine compounds according with time of contact. For this approach, one matrix was built based on the common volatile compounds to white and red wine within all putatively identified compounds. Biplots in PC1 versus PC2 plane combining scores plot and loading plot was constructed using autoscaled data through Metaboanalyst 4.0 (web software, The Metabolomics Innovation Centre (TMIC), Edmonton, AB, Canada) [30].

Student's test was employed to compare chromatographic peak areas of each volatile compound in the different wines using Microsoft Excel 2019.

### 3. Results and Discussion

To identify the reactions that can be promoted by Ch-Ge films, wine model solutions with sugars and amino acids were prepared. Model solutions of volatile compounds were also prepared to assess their interactions with Ch-Ge films. To evaluate these effects on a real wine matrix, the volatile composition of wines in contact with Ch-Ge films during 2 and 8 months were analysed.

### 3.1. Promotion of Maillard Reactions by Chitosan-Genipin Films in Wine Model Solutions

The wine model solutions of arabinose (Ara) containing Ch-Ge films showed the formation of furfural, which increased along the 120 days (Figure 1A). However, furfural was not detected in the absence of the films neither in the presence of genipin (Ge) and glycerol (Gro) (Table 2). These results may be explained by the reaction of amino group of chitosan with the aldehyde group of Ara resulting in the furfural formation. This is in accordance with previous studies that showed the reaction of chitosan amino groups with monosaccharides [31,32] and reducing disaccharides [31,33], producing brown pigments as final stage products of Maillard reaction. A similar mechanism, where chitosan reacts with fructose at higher temperature and low water activity, explains the mitigation of acrylamide formation by competition of chitosan amino groups with asparagine [34].

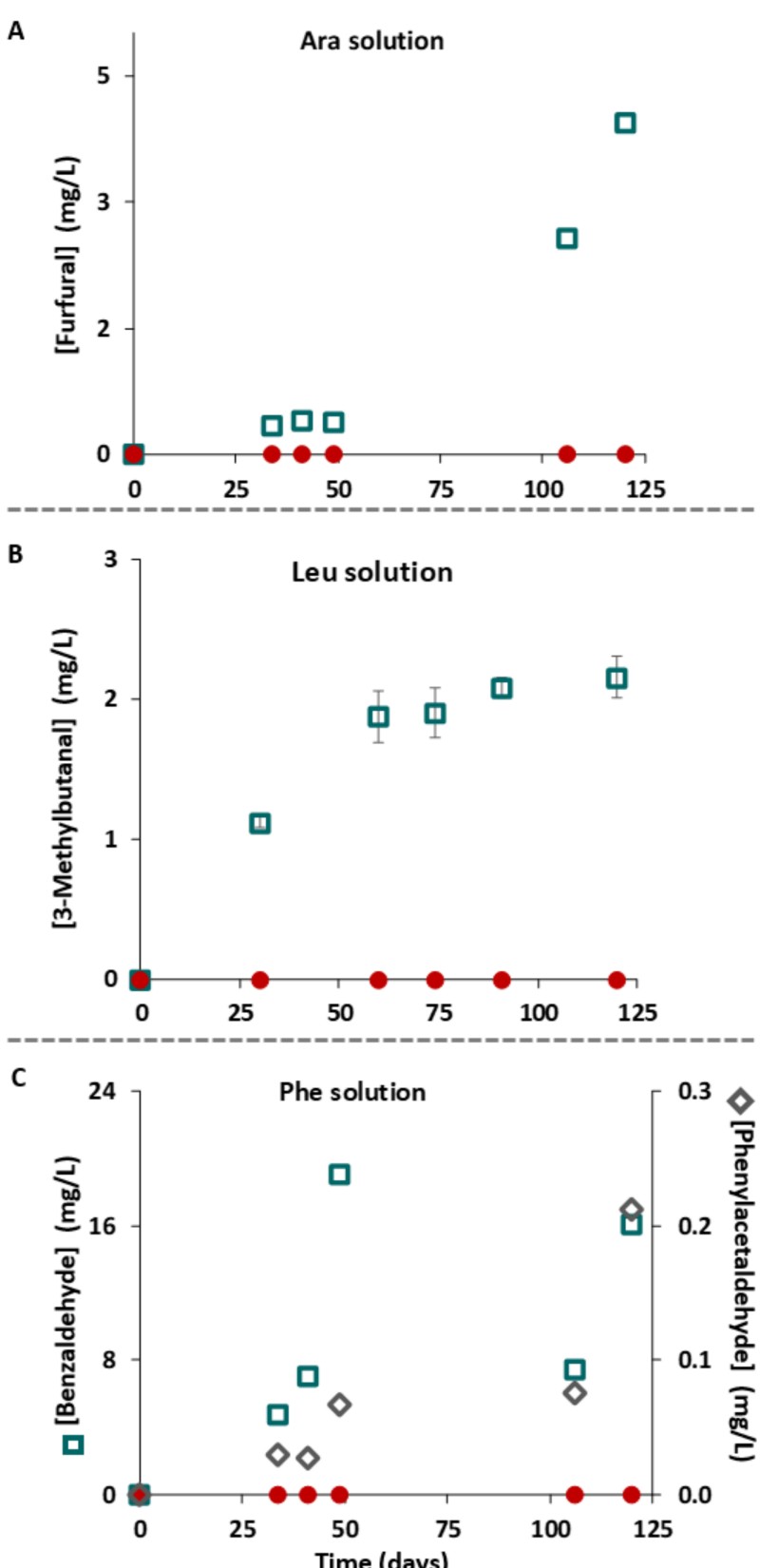

**Figure 1.** Maillard volatile compounds formation in wine model solutions containing (**A**) arabinose (Ara), (**B**) leucine (Leu), and (**C**) phenylalanine (Phe), in the absence (●) and presence of chitosan-genipin films (□, ◇) at room temperature in the dark during 120 days.

**Table 2.** Maillard volatile compounds formation in wine model solution containing arabinose (Ara) or phenylalanine (Phe) in the presence of glycerol (Gro) and genipin (Ge), the components of chitosan-genipin films, at room temperature in the dark during 36 and 120 days.

| Wine Model Solution | Time (Days) | [Maillard Volatile Compounds] (mg/L) | | |
| --- | --- | --- | --- | --- |
| | | Furfural [a] | Phenylacetaldehyde [b] | Benzaldehyde [c] |
| Ara + Gro | 36 | * | * | * |
| | 120 | * | * | * |
| Ara + Ge | 36 | * | * | * |
| | 120 | * | * | * |
| Phe + Gro | 36 | * | * | * |
| | 120 | * | * | * |
| Phe + Ge | 36 | * | $7.5 \times 1.0$ | $0.02 \times 0.01$ |
| | 120 | * | $62.6 \times 6.0$ | $0.18 \times 0.01$ |

* Below limit of detection (LOD): [a] LOD (furfural): 0.089 mg/L; [b] LOD (phenylacetaldehyde): 0.099 mg/L; [c] LOD (benzaldehyde): 0.010 mg/mL.

The wine model solutions of leucine (Leu) containing Ch-Ge films showed the formation of 3-methylbutanal (Figure 1B), possibility resulting from the decarboxylation and oxidative deamination of Leu through a Strecker degradation. The same Strecker degradation mechanism can explain the formation, in wine model solutions containing phenylalanine and Ch-Ge films, of phenylacetaldehyde and benzaldehyde (Figure 1C), the latter explained by the decarboxylation of phenylacetaldehyde [35]. For the formation of amino acid Strecker degradation reactions it is necessary the presence of an aldehyde group. Chitosan has, in fact, a carbonyl group at its reducing terminal. As a chitosan with an average of 150 kDa molecular weight was used to prepare the Ch-Ge films, 2 cm$^2$ of film (Ch, 10 mg) had 0.07 μmol of reducing terminals, a value much lower than the amount of Leu (30 μmol) and Phe (30 μmol) used in these experiments. Although the amount of 3-methylbutanal formed after 120 days (0.08 μmol) can be explained by this hypothesis, the amount of benzaldehyde (0.4 μmol) cannot, as the amount of Strecker aldehydes formed in the wine model solution containing phenylalanine and Ch-Ge films is 6-fold higher than that explained by the amount of this reducing terminal. Therefore, other components of Ch-Ge film, namely genipin (1.0 μmol), which is a dialdehyde, although crosslinked, may still have free aldehyde groups able to be involved in the Strecker degradation of amino acids. To confirm this hypothesis, a wine model solution containing Phe and Ge was analysed. This experiment showed the formation of phenylacetaldehyde and benzaldehyde (Table 2), but not when Phe was in contact with Gro, showing that Ge promotes Strecker degradation. Figure 2 shows the proposed mechanism for the formation of the phenylacetaldehyde and benzaldehyde by Ch-Ge films. The results reported are in accordance with the ability of genipin to react with amino acids for different purposes, namely structural characterisation of products resultant from Ge reaction with glycine [36] and amino acids quantification [37].

### 3.2. Volatile Compounds Retention Capacity of Chitosan-Genipin Films in Wine Model Solutions

To assess if Ch-Ge films can have specific interactions with wine volatile compounds, assays were designed using wine model solutions containing individual representative compounds of carboxylic acids (hexanoic acid), esters (ethyl acetate and ethyl hexanoate), aldehydes (hexanal and benzaldehyde), ketones (octan-3-one), and alcohols (hexanol). These compounds were selected considering their different functional groups with carbon chains ranging from C4 to C8. The content of each volatile compound in the vapour phase, expressed as mg/L, was determined along at least 4 h, using as reference wine model solutions without Ch-Ge films (Figure 3). For ethyl hexanoate and hexanoic acid, experiments were also performed using different proportion of Ch-Ge films for the same volume of solution.

It is possible to observe for ethyl hexanoate and hexanoic acid that their amount decreased along time. After 4 h, the solutions containing 0.5 cm$^2$ of Ch-Ge film presented 17% and 19% less when compared to reference solutions of ethyl hexanoate and hexanoic acid, respectively. This decrease was even higher when 1 cm$^2$ and 2 cm$^2$ were used, 35% for ethyl hexanoate and 45% for hexanoic acid, 38% for ethyl hexanoate and 54% for hexanoic acid, respectively. A decrease in the presence of film was also observed for octan-3-one (28%) using 2 cm$^2$ of film along 4 h. However, this was not observed for ethyl acetate and hexanol. The difference between ethyl hexanoate and ethyl acetate may be due to the higher hydrophobicity of the former (log P, 2.83, Table 1) than the later (log P, 0.73), confirming the promotion of hydrophobic interaction between Ch-Ge films and long chain hydrophobic compounds. This property should also explain the decrease observed for octan-3-one (log P, 2.30) after 4 h.

The higher decrease in wine model solutions of hexanoic acid when compared with hexanol may be related to the negative charges of the acid (pKa, 4.88) that can electrostatically interact with Ch-Ge films positive charges (pKa, 6.5).

The wine model solutions containing hexanal showed a different behaviour from hexanoic acid and, also, from hexanol. It is observed a 12% decrease in the amount of this aldehyde after 4 h. It is possible that hexanal can react with Ch-Ge film amino groups, forming Schiff bases [38] and originating Amadori compounds. The same behaviour, with a decrease of 16%, was observed for benzaldehyde, in accordance with the data obtained for Ch-Ge films wine model solutions containing Ara.

**Figure 2.** Hypothetical pathway leading to formation of phenylacetaldehyde and benzaldehyde from chitosan-genipin films and phenylalanine.

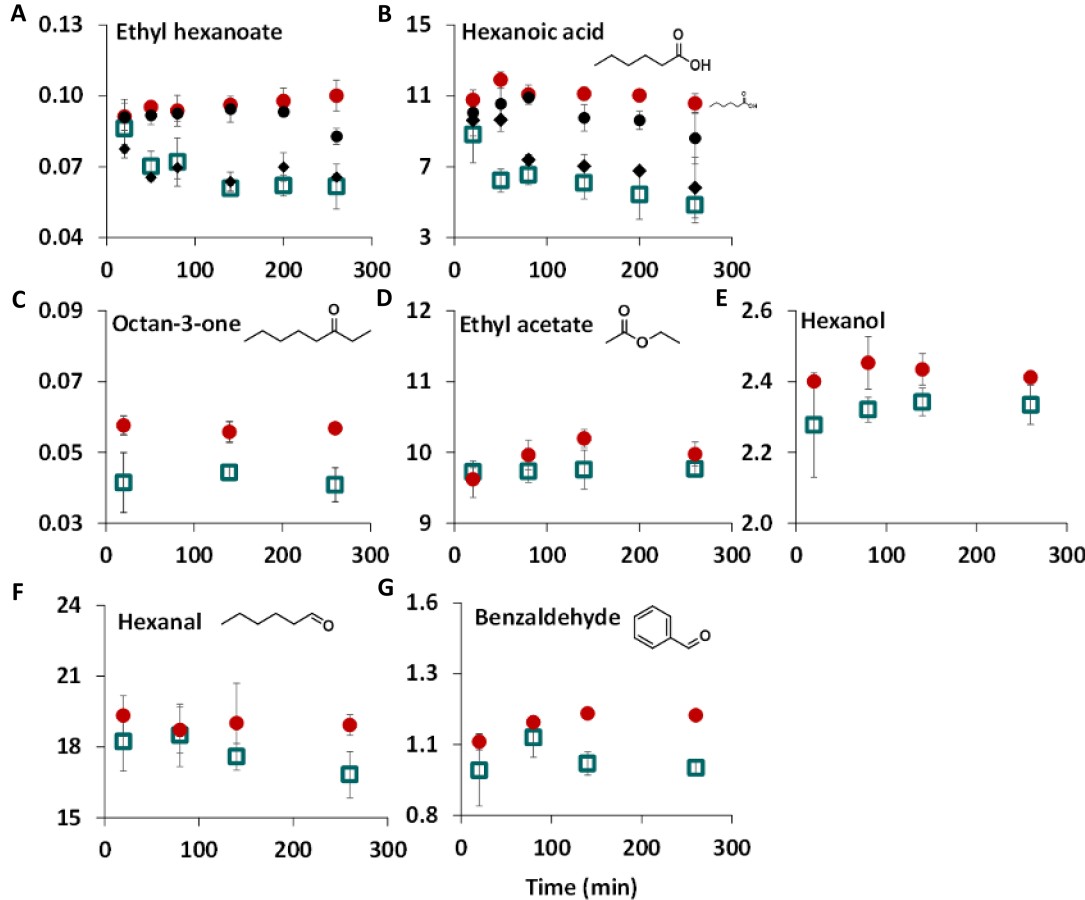

**Figure 3.** Evolution of the volatile compound concentration (mg/L) determined by HS-SPME-GC-FID, in wine model solutions without (●) and with chitosan-genipin films (2 cm² □, 1 cm² ◇, and 0.5 cm² ●) at room temperature in dark during 260 min for (**A**) ethyl hexanoate, (**B**) hexanoic acid, (**C**) octan-3-one, (**D**) ethyl acetate, (**E**) hexanol, (**F**) hexanal, and (**G**) benzaldehyde.

### 3.3. Impact of Chitosan-Genipin Films on Wine Volatile Composition

The volatile composition of white and red wines that were in contact with chitosan-genipin (Ch-Ge) films along 2 and 8 months is represented by two heatmaps, one with the 177 compounds identified in white wines (WW, Figure 4) and another with the 206 compounds identified in red wines (RW, Figure 5). After 8 months, esters, acids, and ketones tend to decrease their concentration in both white and red wines (blue colour), whereas furans and aldehydes tend to increase (red colour). The decrease in ketones (as heptan-2-one and nonan-2-one) and the increase in furans (furfural) and aldehydes (benzaldehyde) were observed in a previous work for white wine treated with Ch-Ge films when compared with untreated wines [21]. Nevertheless, these variations were not observed for all compounds in the same chemical family. The detailed information about the volatile compounds identified by HS-SPME/GC × GC—ToFMS in white and red wines is provided in the Supplementary Materials (Table S1).

Grouping the 139 compounds identified in all wines according to their hydrophobicity index, log P, it is possible to observe that the decrease after 8 months occurred mainly due to the non-polar (log P ≥ 2.3) compounds. The decrease was 1.65-fold lower ($p < 0.05$) for white wine (Figure 6), which corresponded to 32% of the area of the 78 compounds, and 1.87-fold lower ($p < 0.05$) for red wine, corresponding to 15%. Although the total GC × GC peak areas for these 139 compounds were similar for white and red wines, $9.4 \times 10^8$ and $9.1 \times 10^8$, respectively, the non-polar compounds of white wines showed higher GC × GC peak areas than those of red wines, 2.0 times for 2 months and 2.3 times for 8 months.

Nevertheless, the films retention effect was similar in both wines, seeming to be ruled by hydrophobic interactions. Principal component analysis of these datasets (Figure 7) is observed in PC1 (42% of variability) the separation of white (PC1 negative) from red wines (PC1 positive). PC2 (30% of variability) allowed the separation of the wines with longer contact with films (PC2 negative) from those with a shorter contact (PC2 positive). The white wines in contact with the films during 2 months are explained by PC1 negative, with low influence in PC2, whereas the red wines in contact with the film during 2 months are explained by PC2 positive, with low influence in PC1. The loadings plot showed that a large majority of non-polar compounds are represented in PC1 negative and PC2 positive quadrant, evidencing a possible interaction of these compounds with Ch-Ge films along the time of contact. This is even more evident for log P > 4 compounds.

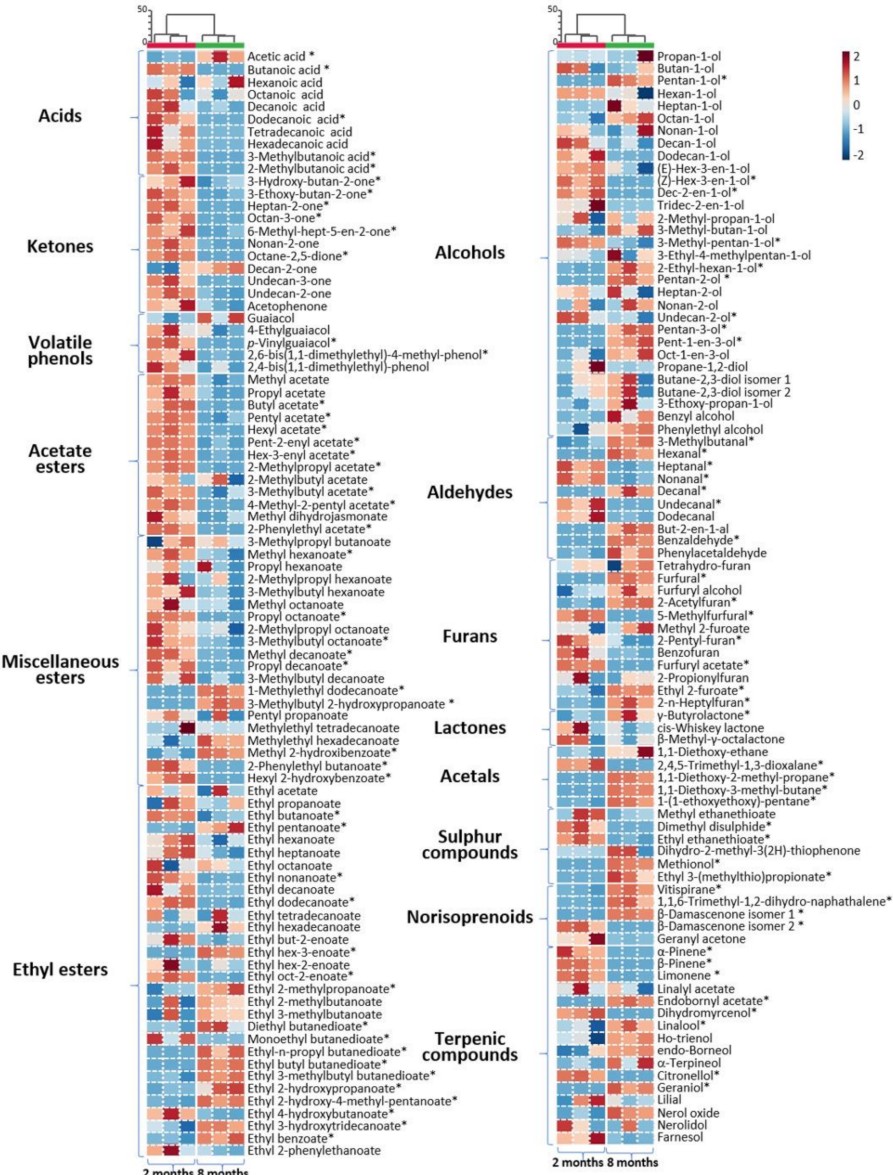

**Figure 4.** Heat map representation corresponding to the 177 volatile compounds of the white wine in contact with chitosan-genipin films during 2 and 8 months. The relative content of each volatile compound is illustrated through a colour scale (from dark blue, minimum, to dark red, maximum). * Significance at $p < 0.05$. Detailed data in Table S1.

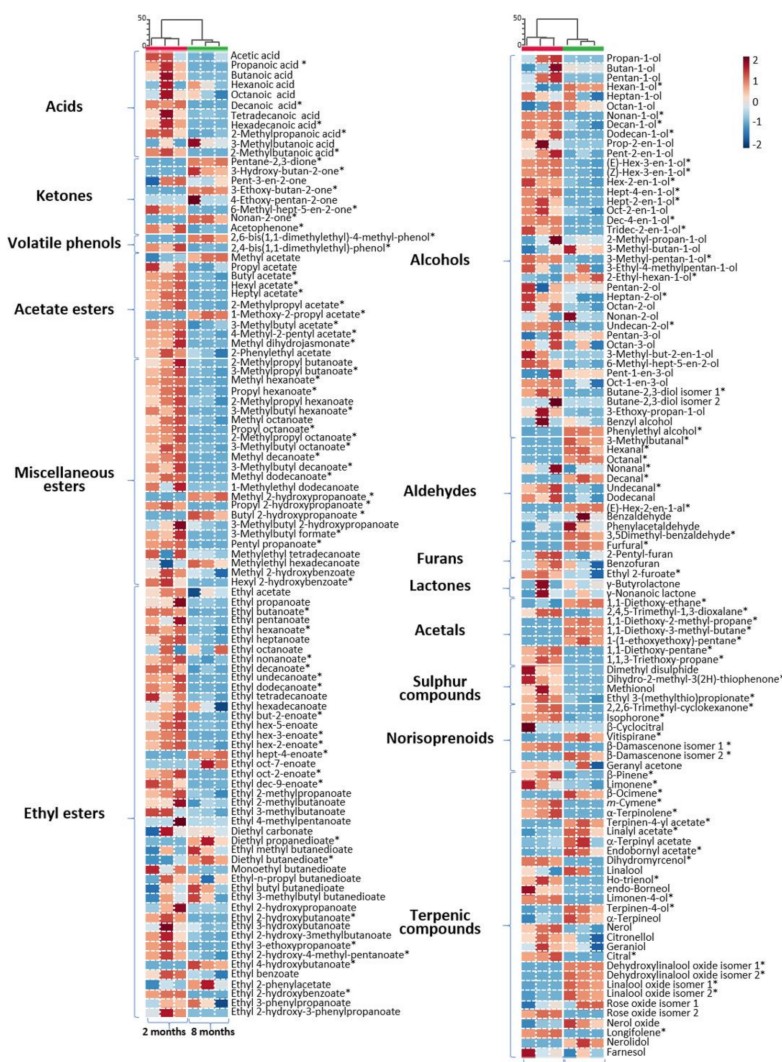

**Figure 5.** Heat map representation corresponding to the 206 volatile compounds of the red wine in contact with chitosan-genipin films during 2 and 8 months. The relative content of each volatile compound is illustrated through a colour scale (from dark blue, minim, um, to dark red, maximum). * Significance at *p* < 0.05. Detail data in Table S1.

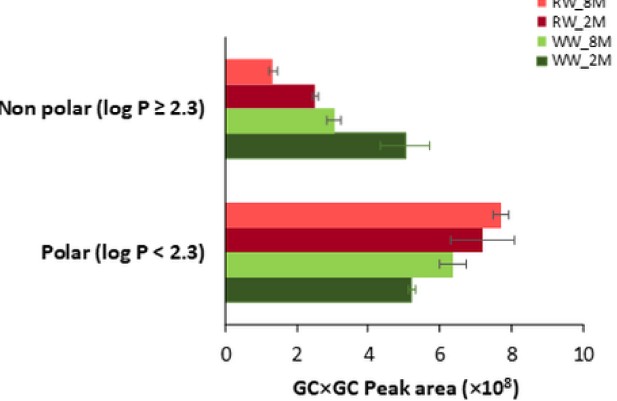

**Figure 6.** GC × GC—ToFMS peak area of the same volatile compounds identified (ωindicated in Table S1) in white wine (WW) and red wine (RW) in contact with chitosan-genipin films during 2 (2 M) and 8 (8 M) months. Log P is the logarithm of the octanol/water partition coefficient of the compounds in this biphasic system.

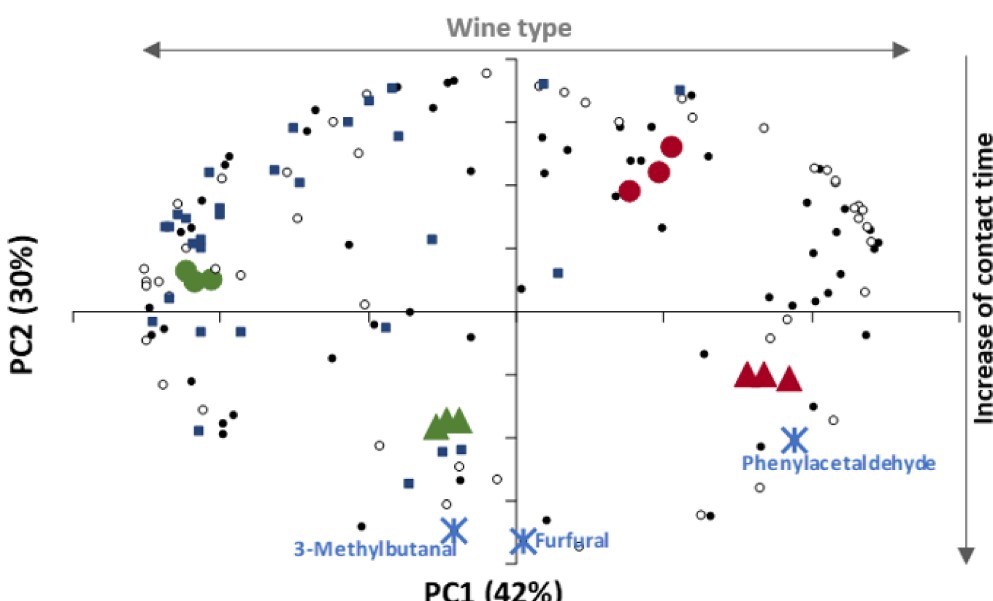

**Figure 7.** Biplots in PC1 × PC2 plane combining scores and loadings plots (● log P < 2.3, ○ 2.3 ≤ log P ≤ 4, and ■ log P > 4) of the white (● and ▲) and red wines (● and ▲) in contact with chitosan-genipin films after 2 (● and ●) and 8 months (▲ and ▲) of storage, related to 139 volatile compounds (indicated in Table S1).

The hydrophobicity effect resulting in the retention of volatile compounds by Ch-Ge films was reflected in both wines volatile composition, namely in ethyl esters family. This group of compounds showed a significant decrease ($p < 0.05$) after 8 months in contact with Ch-Ge films, especially for the long chain esters, such as ethyl nonanoate (in WW and RW), ethyl decanoate (in RW), and ethyl dodecanoate (in WW and RW), as shown in Figures 4 and 5. These results are in accordance with the wine model solutions behaviour for ethyl hexanoate when compared with ethyl acetate (Figure 3). The same behaviour was observed for alcohols, where a significant decrease ($p < 0.05$) of their amount was also observed, with statistically significant differences for decan-1-ol (in RW), undecan-2-ol (in WW and RW), and dodecan-1-ol (in RW), in accordance with the non-retention of hexanol in wine model solutions. The inhibition by chitosan [39] of lipoxygenase and other enzymatic activities still present in wine [40] may explain the lower content of (Z)-hex-3-en-1-ol and 3-methyl-pentan-1-ol in the Ch-Ge wines.

As observed for model solutions containing hexanoic acid (Figure 3), a significant decrease ($p < 0.05$) of the amount of acids after 8 months in both wines in contact with Ch-Ge films was also observed, statistically significant for butanoic (in WW), decanoic (in RW), hexadecanoic (in RW), and 2-methylbutanoic acids (in WW and RW), as shown in Figures 4 and 5.

The principal component analysis performed to the GC × GC peak areas of the volatile compounds of the wines in contact with Ch-Ge films (Figure 7) placed in PC2 negative the wines with longer contact with films due to the furfural, 3-methylbutanal, and phenylacetaldehyde, which increase their content in these wines (Figures 4 and 5, and Table S1). These results are in accordance with the data obtained for the wine model solutions containing pentoses or amino acids in the presence of Ch-Ge films, which formed furfural (Figure 1A) or Strecker aldehydes (Figure 1B,C), respectively. The content of furfural had an increase in the wines in contact with Ch-Ge film during 8 months when compared with wines in contact with the films during 2 months: from $1.7 \times 10^6$ to $8.9 \times 10^6$, $p < 0.05$ for white wines (Table S1 and Figure 4) and from below the limit of detection ($6.0 \times 10^3$) to $4.8 \times 10^6$, $p < 0.05$ for red wines (Table S1 and Figure 5). The occurrence of furfural in wines is usually valorised due to its association with toasted sweet foods [41], giving

particular notes to aged white wines in toasted wood barrels [42,43]. Phenylacetaldehyde, benzaldehyde, and 3-methylbutanal, Strecker degradation derived products, known to confer positive distinctive notes to wine aroma, should provide fruity [44], almond [45], and malty [46] aroma scents to wines in contact with Ch-Ge films.

Touriga Nacional wines are characterised by their content on monoterpenic compounds, such as linalool, $\alpha$-terpineol, and geraniol [47,48]. The amount of these compounds did not show significant differences in the presence of Ch-Ge films. For Encruzado, no data are available concerning its varietal aroma characteristics [27,49]. Nevertheless, the chemical families responsible for the varietal aroma in white wines are monoterpenic, sesquiterpenic, and norisoprenic compounds. It is possible to observe that Ch-Ge films had low impact on these classes in white and red wines (Figures 4 and 5), allowing to explain why the white wines were highly appreciated by the panel test of experts [21].

## 4. Conclusions

Chitosan crosslinked with genipin (Ch-Ge) films, when immersed in wines, modify their volatile composition through the formation and retention of compounds. Ch-Ge films, through chitosan primary amino groups, can react with reducing sugars present in wine, through Amadori reactions, allowing the formation of furan derivatives. It was shown in this study that pentoses, in presence of Ch-Ge films, formed furfural in proportions able to provide toasted aroma notes to the wines. In addition, Ch-Ge films, through free aldehydes available in genipin and reducing terminal of chitosan, can form Strecker aldehydes by reaction with amino acids. Phenylacetaldehyde, benzaldehyde, and 3-methylbutanal may provide positive distinctive fruity, green, floral, honey, and bitter almond aroma notes to these wines. Ch-Ge films retention capacity increases with the hydrophobicity of the compound for the same chemical family. For the same carbon chain of volatile compounds, the acids were more retained by Ch-Ge films than alcohols. This work showed that the use of Ch-Ge films opens new opportunities for winemaking industries as technological adjuvant, acting as preservative and also positively modulating wine volatile compounds without changing varietal aroma.

**Supplementary Materials:** The following are available online at https://www.mdpi.com/article/10.3390/app11146294/s1, Table S1: Area of volatile compounds identified by HS-SPME/ GC × GC-ToFMS in white and red wines in contact with chitosan-genipin films during 2 and 8 months.

**Author Contributions:** Conceptualisation, M.A.C. and C.N.; methodology, M.A.M.R. and S.M.R.; formal analysis, M.A.M.R.; writing—original draft preparation, M.A.M.R.; writing—review and editing, S.M.R., M.A.C. and C.N.; supervision, M.A.C. and C.N. All authors have read and agreed to the published version of the manuscript.

**Funding:** This research was also developed within the scope of the project CICECO-Aveiro Institute of Materials (UIDB/50011/2020 and UIDP/50011/2020), QOPNA (UID/QUI/00062/2019) and LAQV-REQUIMTE (UIDB/50006/2020), financed by national funds through the FCT/MEC and when appropriate co-financed by FEDER under the PT2020 Partnership Agreement. M. Angélica M. Rocha and Cláudia Nunes thank FCT for the Doctoral (SFRH/BD/98015/2013) and Post-Doctoral fellowships (SFRH/BPD/100627/2014), respectively. This work was also funded by national funds (OE), through FCT, I.P., within the scope of the framework contract foreseen in the numbers 4, 5, and 6 of the article 23, of the Decree-Law 57/2016, of 29 August, changed by Law 57/2017, of 19 July.

**Institutional Review Board Statement:** Not applicable.

**Informed Consent Statement:** Not applicable.

**Data Availability Statement:** Not applicable.

**Conflicts of Interest:** The authors declare no conflict of interest.

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
