# Peer review of "Impact of Chitosan-Genipin Films on Volatile Profile of Wine along Storage"

_applsci, doi:10.3390/app11146294_

Round 1

Reviewer 1 Report

L39 and 42: please use SI units here.

The introduction seems too short for me. There is a lack of information on the application of Ch-Ge films, and it would be great to know how other researchers measure these aspects.

Please add more literature to the introduction.

MAtrials and methods.

This section is well written and easy to follow. I would suggest to unify the use of % sign. Eighter put a space between the number and % or not but be concise.

I miss the data analysis part. As the results section introduces cluster analysis and heatmaps (Figure 5), I think the authors should describe it in the methods section.

Resutls:

L344: What test was used here?

PCA: Please add what software was used and give some details regarding the pca. KMO, Bartlett’s test, etc is missing as well as the description of the data set (pretreatments, standardization, missing data handling, etc).

Conclusions

The paper is well-written and the statements seem valid, however the statistical analysis part should be strengthen as well.

Author Response

Comment 1:

L39 and 42: please use SI units here.

REPLY: The units were changed to g/L.

Comment 2:

The introduction seems too short for me. There is a lack of information on the application of Ch- Ge films, and it would be great to know how other researchers measure these aspects. Please add more literature to the introduction.

REPLY: Thanks for the suggestion. The new references recently available in the literature, mainly related with the application of chitosan to wines, have been included. Concerning chitosan-genipin films in wine, all references available have been cited.

Comment 3:

Materials and methods.

This section is well written and easy to follow. I would suggest to unify the use of % sign. Eighter put a space between the number and % or not but be concise.

REPLY: Thanks for the suggestion.

Comment 4:

I miss the data analysis part. As the results section introduces cluster analysis and heatmaps (Figure 5), I think the authors should describe it in the methods section.

REPLY: We are grateful for your observation. This was an omission during the filling of the journal layout. The new section was included in the materials and methods.

Comment 5:

Resutls:

L344: What test was used here?

PCA: Please add what software was used and give some details regarding the pca. KMO, Bartlett’s test, etc is missing as well as the description of the data set (pretreatments, standardization, missing data handling, etc).

Conclusions

The paper is well-written and the statements seem valid, however the statistical analysis part should be strengthen as well.

REPLY: The information about the software used and statistical analysis was included in the new section entitled by Data analysis in the material and methods.

Reviewer 2 Report

Dear authors,

here are some suggestions for corrections in your study:

Line 50- can you explain what does  ´global´ stands for?

75/76 - missing word and bracket

131/132 - Vitis vinifera- in italic

134/144 - et al.

126-128 - does the description of the table refers to its content?

Figure 3- could be better in resolution

References in lines 489, 496, 514, and 531 need to be corrected

After this minor correction, I have noticed that there is a much bigger issue in the experimental design. There is a lack of control wine to be compared with the wines in the experiment. Wines that were used in the experiment were produced during vintage 2010- 11 years ago, and they were already part of your previous study published in 2016. I have an issue with so small sampling especially if they were already a part of another experiment.

Some methods and materials are the same as in the previous study, so it would be good to do it again with some better experiment design- on a larger scale with some recent/future vintage of chosen varieties.

Author Response

here are some suggestions for corrections in your study:

Line 50- can you explain what does  ´global´ stands for?

REPLY: The sentence was rephrased to be clear.

75/76 - missing word and bracket

131/132 - Vitis vinifera- in italic

134/144 - et al.

REPLY: Thanks for the suggestions.

126-128 - does the description of the table refers to its content?

REPLY: We are sorry for this copy and paste mistake when converting the written manuscript to the requested journal layout. The table title has been changed.

Figure 3- could be better in resolution

REPLY: It was changed the resolution of the figure 3

References in lines 489, 496, 514, and 531 need to be corrected

REPLY: The references were corrected

After this minor correction, I have noticed that there is a much bigger issue in the experimental design. There is a lack of control wine to be compared with the wines in the experiment. Wines that were used in the experiment were produced during vintage 2010- 11 years ago, and they were already part of your previous study published in 2016. I have an issue with so small sampling especially if they were already a part of another experiment.

Some methods and materials are the same as in the previous study, so it would be good to do it again with some better experiment design- on a larger scale with some recent/future vintage of chosen varieties.

REPLY: The focus of this work was to demonstrate that chitosan-genipin (Ch-Ge) films had the promotion capacity of Maillard reactions and the interaction with hydrophobic volatile compounds. For this, wine model solutions were used, and the data obtained were compared with previously analyzed wines with results that have not been published. The work with model solutions is not dependent of the vintage, the reason why no wines have been prepared for this purpose.

Reviewer 3 Report

The present work identifies the type of reactions and interactions that can be promoted by Ch-Ge films with impact on the volatile composition of white and red wines in contact with the films during more than 2 months. The use of Ch-Ge films opens new opportunities for winemaking industries as technological adjuvant, for white wine preservation to replace sulfur dioxide, and positively modulating wine volatile compounds without changing varietal aroma.

The paper provides an interesting, original, and winemaking usefully research work.

There are a lot of experiments with apparatus and procedures used and are well described and moreover the results are well explained with adequate discussion.

I have a few remarks:

Results and discussion

In figure 1 the time’s axe must be in days not in hours, as is specified:” Figure 1. Maillard volatile compounds formation in wine model solutions containing … during 120 days”.

References

At reference 25 must delete the signs ????.

I suggest that the author to include more recently references.

Author Response

The present work identifies the type of reactions and interactions that can be promoted by Ch-Ge films with impact on the volatile composition of white and red wines in contact with the films during more than 2 months. The use of Ch-Ge films opens new opportunities for winemaking industries as technological adjuvant, for white wine preservation to replace sulfur dioxide, and positively modulating wine volatile compounds without changing varietal aroma.

The paper provides an interesting, original, and winemaking usefully research work.

There are a lot of experiments with apparatus and procedures used and are well described and moreover the results are well explained with adequate discussion.

I have a few remarks:

Results and discussion

In figure 1 the time’s axe must be in days not in hours, as is specified:” Figure 1. Maillard volatile compounds formation in wine model solutions containing … during 120 days”.

REPLY: The figure was corrected.

References

At reference 25 must delete the signs ????.

I suggest that the author to include more recently references.

REPLY: The reference 25 was revised. The new references recently available in the literature, mainly related with the application of chitosan to wines, have been included.

Round 2

Reviewer 2 Report

After this minor correction, I have noticed that there is a much bigger issue in the experimental design. There is a lack of control wine to be compared with the wines in the experiment. Wines that were used in the experiment were produced during vintage 2010- 11 years ago, and they were already part of your previous study published in 2016. I have an issue with so small sampling especially if they were already a part of another experiment.

Some methods and materials are the same as in the previous study, so it would be good to do it again with some better experiment design- on a larger scale with some recent/future vintage of chosen varieties.

Author Response

This work has a wider data set, where the control wines were also analyzed, namely wines with sulphur dioxide and without any treatment. As stated by the reviewer, the comparison between the volatile composition of control wines and the wine with chitosan-genipin films was described in a previous published work (Green Chem. 2016, 18, 5331–5341, doi:10.1039/C6GC01621A), since this work was focused on the use of the films for wine preservation. However, the aim of this manuscript, that we intend to publish in Applied Sciences, is to identify the type of reactions and interactions that can be promoted by Ch-Ge films to explain the impact on the volatile composition. For accomplish this aim, wine model solutions containing Ch-Ge films were used and the volatile composition of wines with films was analyzed in two contact times (2 and 8 months) to confirm the results obtained with the model solutions. The authors think that analyze more wines are not going to change the conclusions of the work or improve the scientific knowledge.

Round 3

Reviewer 2 Report

Dear authors,

I agree with your statement, but I can not agree with publishing the study on this scientific level by using the 10-years-old wine samples that were already used for another publication,

best regards

Author Response

The wines analysed in this manuscript were part of a set of white and red wines prepared to study the feasibility of Ch-Ge films  (Green Chem. 2016, 18, 5331–5341) and High Hydrostatic Pressure (Food Chem. 2015, 188, 406–414) to replace sulphur dioxide as wine preservative after fermentation. These wines were never used in any previous publication. From the 4 wines under study, one was used in reference from Green Chem. 2016, 18, 5331–5341 (white wine in contact with Ch-Ge films during 8 months). However, in this previous work only the overall volatile composition was presented, not the detailed volatile composition used in this work to compare white and red wines in contact with Ch-Ge films during 2 and 8 months. These results allowed to validate the results obtained with the model solutions, namely the promotion of Maillard reactions and the interaction with hydrophobic volatile compounds. Therefore, as suggested by the reviewer these aspects have been clarified in the revised manuscript.